# Analysing Extrapolation Capabilities of Modern Positional Embeddings in Vision Transformers

## Abstract

Vision Transformers scale quadratically with input resolution, making high-resolution training prohibitively expensive, yet detection and segmentation demand it. A natural alternative is resolution extrapolation: train at low resolution and deploy at higher resolution without fine-tuning. Whether this works depends entirely on the positional encoding. Modern encodings such as RoPE, ALiBi, YaRN, and FIRE enable striking length extrapolation in language models, but their behavior on 2D image grids is largely unknown. We present a systematic study, training ViT-Tiny models at $224 \times 224$ (ImageNet-100) and $64 \times 64$ (Tiny-ImageNet) and evaluating zero-shot up to $1024 \times 1024$ ($4.57\times$) and $512 \times 512$ ($8\times$) on classification, COCO detection, and ADE20K segmentation. Across all configurations, additive attention-bias methods consistently outperform rotation-based methods at large scales: at $4.57\times$, FIRE retains 63.1% Top-1 accuracy while RoPE collapses to 18.9%. A scale-free attention concentration analysis traces this gap to a fundamental mechanistic difference: bias-based encodings actively concentrate attention on a small subset of tokens as resolution grows, whereas rotation-based methods leave attention broadly diffuse across the enlarged grid. We additionally expose a topological interpolation failure in ALiBi and highlight the base-resolution limits of YaRN inference scaling. These results recast extrapolation robustness as the primary axis for choosing positional encodings.

## 1 Introduction

Vision Transformers (ViTs) Dosovitskiy et al. (2021) have emerged as a dominant architecture across computer vision, demonstrating strong performance in both recognition and dense prediction settings. However, ViTs suffer from quadratic computational complexity with respect to token count, making training at high resolutions prohibitively expensive. This limitation is particularly problematic for detection and segmentation, where fine-grained spatial detail is essential. Resolution extrapolation, i.e., training at low resolution and deploying at higher resolution, offers an appealing alternative, but is fundamentally constrained by the poor generalization of standard absolute positional embeddings beyond their training grid.

The practical importance of resolution extrapolation cannot be overstated. Object detection and semantic segmentation require precise localization of fine-grained structures, small objects, and thin boundaries, all of which benefit from higher-resolution input. Yet training a ViT-Base at $1024 \times 1024$ requires roughly $21\times$ the FLOPs compared to $224 \times 224$, making large-scale training at high resolution impractical for many research groups and deployment scenarios. If a model trained cheaply at low resolution could be deployed at higher resolution with minimal accuracy loss, it would dramatically reduce both the computational cost and the carbon footprint of modern vision systems. The critical bottleneck in realizing this goal is the positional encoding: while the feature extraction layers of a ViT are inherently resolution-agnostic (each patch is processed independently before attention), the positional encoding ties the model to a specific spatial grid and determines whether the learned spatial relationships generalize to unseen resolutions.

Recent progress in Large Language Models (LLMs) Touvron et al. (2023) has produced a rich landscape of positional encoding strategies specifically designed for length extrapolation. These methods fall into two broad families. *Rotation-based methods*, such as RoPE Su et al. (2024) and its extension YaRN Peng et al.

(2024), encode relative positions by applying position-dependent complex rotations to query and key vectors, so that the attention score between two tokens depends only on their relative displacement. *Additive bias methods*, such as ALiBi Press et al. (2022) and FIRE Li et al. (2024), instead inject positional information as a deterministic or learned bias added directly to attention logits, leaving the content-based query–key similarity untouched. Both families have enabled dramatic context-length extension in 1D language models, but their behavior on the two-dimensional spatial lattice of images, where relative offsets become 2D vectors and distances grow radially rather than linearly, remains largely unexplored.

A handful of prior works have adapted individual methods to vision, most notably 2D RoPE Heo et al. (2024), but no study has systematically compared rotation-based and bias-based families under controlled conditions across multiple downstream tasks. This gap is significant because the structural properties that make a positional encoding extrapolatable in 1D (translation invariance, monotonic attention decay, suppression of high-frequency oscillations) may not straightforwardly transfer to 2D grids, where the topology, distance metric, and boundary conditions all differ.

In this work, we conduct a systematic empirical study of modern positional encoding strategies adapted to 2D Vision Transformers. We focus on ViT-Tiny models trained on ImageNet-100 Deng et al. (2009) at $224 \times 224$ and Tiny-ImageNet at $64 \times 64$, evaluating zero-shot extrapolation up to $1024 \times 1024$ ($4.57\times$ scaling) and $512 \times 512$ ($8\times$ scaling) respectively without fine-tuning. We evaluate each method on classification, COCO object detection, and ADE20K semantic segmentation, and complement accuracy metrics with a scale-free attention concentration analysis that reveals *why* certain methods extrapolate successfully. Our core contributions are:

- We provide the first comprehensive comparison of RoPE Su et al. (2024); Heo et al. (2024), ALiBi Press et al. (2022), YaRN inference-scaling Peng et al. (2024), and FIRE Li et al. (2024) for 2D resolution extrapolation in Vision Transformers across classification and dense prediction tasks.

- We demonstrate that attention bias-based methods (ALiBi, FIRE) consistently outperform rotation-based methods (RoPE, YaRN) under extreme extrapolation, a finding we rigorously validate across multiple training regimes and datasets.

- We introduce a resolution-comparable attention concentration metric, proving mathematically that bias-based methods actively concentrate attention as spatial grids grow, whereas rotation-based methods fail to prevent diffuse attention spread.

- We ablate ALiBi's catastrophic failure at interpolated resolutions, proving it stems from a fundamental topological inability to adapt to grid size reductions, and identify base-resolution constraints that restrict YaRN's applicability.

## 2 Related Works

### 2.1 Positional Encoding for Transformers

Transformers Vaswani et al. (2017) are permutation-invariant and require explicit positional information to model token order or spatial layout. Early methods employ absolute positional embeddings, either fixed sinusoidal or learned, added directly to token features. Vision Transformers Dosovitskiy et al. (2021) adopt learned 2D absolute embeddings defined on a fixed spatial grid, using bicubic interpolation Beyer et al. (2022) when transferring to higher resolutions. However, such embeddings are resolution-specific and do not provide true zero-shot extrapolation.

Relative positional embeddings (RPEs) instead encode pairwise displacements by modifying attention scores, ensuring translation invariance and enabling generalization to unseen positions. In NLP, methods such as those of Shaw *et al.* Shaw et al. (2018) and T5 Raffel et al. (2020) improve sequence-length generalization, while in vision, Swin Transformer Liu et al. (2021) introduces learnable 2D relative position biases within local windows. More recent approaches such as FlexiViT Beyer et al. (2023) and conditional positional encodings Chu et al. (2023) explore resizing strategies for both patch and positional embeddings, but still

rely on bounded lookup tables or learned absolute embeddings with limited support for long-range spatial extrapolation.

The challenge of length extrapolation has been most actively studied in large language models. Several key properties have been identified for successful extrapolation Hong et al. (2024); Dong et al. (2024): translation-invariant dependence on relative offsets, monotonic or smoothly decaying attention with distance, and suppression of high-frequency oscillations that destabilize long-range correlations. Building on these principles, a range of methods have been proposed, including RoPE Su et al. (2024), ALiBi Press et al. (2022), YaRN Peng et al. (2024), FIRE Li et al. (2024), position interpolation Chen et al. (2024), XPOS Sun et al. (2022), InfLLM Xiao et al. (2024), divide-and-conquer scaling Yang et al. (2025), mixed-radix extensions Tian et al. (2026), imaginary RoPE extensions Liu et al. (2025), multi-scale self-injection Han et al. (2026), and context window scheduling Zhu et al. (2025). While these mechanisms have enabled dramatic context extension in 1D, existing analyses are almost exclusively confined to linear token sequences; how these extrapolation principles transfer to 2D spatial grids, where offsets become vectors and distances grow radially, remains largely unexplored.

## 2.2 Positional Embeddings in Vision

In the visual domain, the adaptation of modern positional encodings is still in its early stages. The original ViT and subsequent works Dosovitskiy et al. (2021); Beyer et al. (2022) interpolate absolute positional embeddings at new resolutions, enabling fine-tuning but not zero-shot extrapolation. Among relative methods, 2D RoPE Heo et al. (2024) has been applied to vision transformers by extending rotary embeddings to two spatial axes, but has not been systematically compared against other modern encodings. ALiBi, FIRE, and YaRN, despite their proven 1D extrapolation capabilities, have received little attention in the vision literature.

This gap is our primary motivation. Existing studies evaluate individual methods in isolation or focus exclusively on classification; no prior work provides a controlled comparison of rotation-based and bias-based positional encodings across classification, detection, and segmentation under zero-shot resolution extrapolation. Our study fills this gap by benchmarking five representative methods under identical training and evaluation conditions.

# 3 Methodology

This section describes each positional encoding strategy evaluated in our study. We begin with the baseline Vision Transformer using learned absolute positional embeddings, then present the rotation-based and bias-based relative positional methods adapted for 2D spatial inputs.

## 3.1 Simple ViT: Learned Absolute Positional Embeddings

The standard Vision Transformer (ViT) Dosovitskiy et al. (2021); Beyer et al. (2022) processes an image $\mathbf{x} \in \mathbb{R}^{H \times W \times C}$ by partitioning it into a grid of non-overlapping patches of size $P \times P$, yielding $N = HW/P^2$ patch tokens. Each patch is linearly projected to a $d$-dimensional embedding via a convolutional stem, and a learnable classification token $\mathbf{x}_{\mathrm{cls}} \in \mathbb{R}^d$ is prepended to the sequence. Positional information is injected by adding a set of learnable absolute positional embeddings $\mathbf{E}_{\mathrm{pos}} \in \mathbb{R}^{N \times d}$ to the patch tokens:

$$\mathbf{z}_0 = [\mathbf{x}_{\mathrm{cls}}; \mathbf{x}_1\mathbf{E}; \mathbf{x}_2\mathbf{E}; \cdots; \mathbf{x}_N\mathbf{E}] + \mathbf{E}_{\mathrm{pos}}, \tag{1}$$

where $\mathbf{E} \in \mathbb{R}^{P^2 C \times d}$ is the patch projection matrix. The resulting sequence is passed through $L$ transformer encoder blocks, each consisting of multi-head self-attention (MSA) and an MLP with layer normalization and residual connections:

$$\mathbf{z}'_\ell = \mathrm{MSA}(\mathrm{LN}(\mathbf{z}_{\ell-1})) + \mathbf{z}_{\ell-1}, \tag{2}$$

$$\mathbf{z}_\ell = \mathrm{MLP}(\mathrm{LN}(\mathbf{z}'_\ell)) + \mathbf{z}'_\ell. \tag{3}$$

The final representation of the classification token $\mathbf{z}_0^L$ is used for prediction.

Since the learned positional embeddings are defined on a fixed $H/P \times W/P$ grid, they cannot directly generalize to spatial grids of different sizes. When evaluating at a resolution different from training, we apply 2D bicubic interpolation to the positional embeddings, reshaping them to the original grid, interpolating to the target grid size, and flattening back to a sequence. While this enables inference at novel resolutions, the interpolated embeddings are only an approximation and may degrade as the resolution gap increases.

### 3.2 RoPE: Rotary Position Embeddings for 2D Vision

Rotary Position Embedding (RoPE) Su et al. (2024) encodes positional information by applying position-dependent complex rotations directly to the query and key vectors in self-attention, rather than adding an explicit positional embedding to the input tokens. For a token at 1D position $n$, RoPE defines a rotation matrix using frequency $\theta_t$ for the $t$-th dimension pair:

$$\mathbf{R}(n,t) = e^{i\theta_t n}, \quad \theta_t = \theta_{\text{base}}^{-t/(d_{\text{head}}/2)}, \tag{4}$$

where $d_{\text{head}}$ is the per-head dimension and $\theta_{\text{base}}$ controls the frequency range. The rotation is applied to query and key vectors via the Hadamard product:

$$\bar{\mathbf{q}}'_n = \bar{\mathbf{q}}_n \circ \mathbf{R}(n), \quad \bar{\mathbf{k}}'_m = \bar{\mathbf{k}}_m \circ \mathbf{R}(m), \tag{5}$$

where $\bar{\mathbf{q}}, \bar{\mathbf{k}} \in \mathbb{C}^{d_{\text{head}}/2}$ are complex-valued views obtained by pairing consecutive dimensions. The resulting attention score between positions $n$ and $m$ naturally encodes their relative displacement:

$$A'_{(n,m)} = \text{Re}[\bar{\mathbf{q}}_n \bar{\mathbf{k}}_m^* e^{i\theta_t(n-m)}]. \tag{6}$$

To adapt RoPE to the 2D spatial structure of Vision Transformers, we follow the mixed learnable frequency formulation Heo et al. (2024). Each patch has a 2D position $\mathbf{p}_n = (p_n^x, p_n^y)$ on the spatial grid. The rotation matrix is defined using separate frequency vectors for each axis:

$$\mathbf{R}(n,t) = e^{i(\theta_t^x p_n^x + \theta_t^y p_n^y)}, \tag{7}$$

where $(\theta_t^x, \theta_t^y)$ are per-head, per-layer learnable frequency parameters. Unlike axial RoPE, which restricts each frequency dimension to a single spatial axis, the mixed formulation allows each frequency to attend to both horizontal and vertical directions simultaneously. This enables RoPE to capture diagonal spatial relationships, which axial frequencies cannot represent. The resulting attention matrix encodes relative 2D displacements:

$$A'_{(n,m)} = \text{Re}\left[\bar{\mathbf{q}}_n \bar{\mathbf{k}}_m^* e^{i(\theta_t^x(p_n^x - p_m^x) + \theta_t^y(p_n^y - p_m^y))}\right]. \tag{8}$$

In our implementation, the mixed frequencies are initialized from a base frequency schedule with random per-head axis rotations and are trained jointly with the network parameters. Rotary embeddings are applied to all patch tokens but skipped for the classification token, which has no spatial position. For detection and segmentation backbones that omit the classification token, RoPE is applied to all tokens directly. Since RoPE does not introduce fixed-size positional parameters, it naturally handles variable-length sequences; however, at resolutions beyond the training range, the effective phase angles move outside the distribution seen during training, which can degrade attention stability.

### 3.3 FIRE: Functional Interpolation for Relative Positional Encoding

FIRE (Functional Interpolation for Relative Positional Encoding) Li et al. (2024) models relative positional information as a continuous, learnable attention bias designed for length extrapolation. Instead of relying on discrete lookup tables, FIRE parameterizes the attention bias between a query position $i$ and a key position $j$ using a small MLP:

$$b(i,j) = f_\theta\left(\frac{\psi(|i-j|)}{\psi(\max\{L,i\})}\right), \tag{9}$$

where $f_\theta : \mathbb{R} \to \mathbb{R}^H$ outputs head-wise biases, $\psi(x) = \log(1 + cx)$ is a monotonic concave transform with learnable scale $c$, and $L$ is a reference length controlling the onset of extrapolation. The normalization by $\max\{L, i\}$ implements *progressive interpolation*, constraining the MLP input to $[0, 1]$ for arbitrary sequence lengths and ensuring that longer contexts correspond to interpolation rather than out-of-distribution extrapolation in function space.

**Adapting FIRE to 2D image grids.** FIRE was originally formulated by Li et al. for one-dimensional token sequences in large language models, where the relative offset $|i - j|$ is a scalar and the per-position normalizer $\psi(\max\{L, i\})$ depends on a single coordinate. Three structural properties make 1D FIRE extrapolatable to unseen sequence lengths: (a) the log warping $\psi(x) = \log(1 + cx)$ allocates higher attention resolution to short distances and dampens far-context contributions; (b) the normalizer $\psi(\max\{L, i\})$ implements *progressive interpolation* by bounding the MLP input to a fixed range $[0, 1]$ regardless of sequence length, so the MLP never sees out-of-distribution inputs at extrapolation; and (c) the per-head MLP $f_\theta : \mathbb{R} \to \mathbb{R}^H$ converts this normalized distance into smooth, learnable head-wise biases. An *appropriate* 2D extension of FIRE is therefore one that preserves all three of these properties on a spatial grid; if any of them is broken, the original justification for FIRE's extrapolation behaviour no longer applies in 2D.

Two seemingly natural 2D extensions fail this test, and it is worth ruling them out explicitly. A *flattened* variant that reuses $|i - j|$ on the linear patch index destroys 2D spatial geometry: two patches on the same image row are related identically to two patches in different rows, removing the very locality structure that the bias is meant to encode, and breaking property (a) in the sense that "distance" no longer corresponds to true spatial distance. A *Euclidean* variant, $b(i, j) = f_\theta(\psi(\|\mathbf{p}_i - \mathbf{p}_j\|_2) / \psi(\max\{L, ?\}))$, has a different problem: no single scalar query-side quantity on a 2D grid plays the role of FIRE's $\max\{L, i\}$, so any Euclidean version must invent an ad-hoc 2D normalizer with no 1D counterpart, and property (b) — the progressive-interpolation guarantee that underpins FIRE's extrapolation behaviour — is no longer inherited from 1D FIRE.

We instead adopt an *axis-decomposable* 2D extension that preserves FIRE's three load-bearing properties — logarithmic warping, progressive interpolation, and query-position-indexed normalization — symbol-for-symbol along each spatial axis. For an $H_p \times W_p$ patch grid, each patch $i$ has 2D coordinates $(x_i, y_i)$ with $x_i = i \bmod W_p$ and $y_i = \lfloor i/W_p \rfloor$. We apply 1D FIRE independently to the $x$- and $y$-axes and sum the two normalized terms before the per-head MLP:

$$b(i, j) \; = \; s \cdot f_\theta\left( \frac{\psi(|x_i - x_j|)}{\psi(\max\{L,\, x_i\})} \; + \; \frac{\psi(|y_i - y_j|)}{\psi(\max\{L,\, y_i\})} \right), \tag{10}$$

where the per-head MLP $f_\theta : \mathbb{R} \to \mathbb{R}^H$, the learnable warping scale $c$ inside $\psi$, and the learnable threshold $L$ are shared across both axes, and $s$ is a global bias scale (described below). The resulting bias tensor $\mathbf{B} \in \mathbb{R}^{H \times N \times N}$ is shared across all transformer layers and added to the scaled dot-product attention:

$$\mathrm{Attn}(Q, K, V) \; = \; \mathrm{softmax}\left( \frac{QK^\top}{\sqrt{d}} + \mathbf{B} \right) V. \tag{11}$$

**Why axis-decomposition is the appropriate 2D extension of FIRE.** Equation 10 preserves all three of FIRE's structural properties on the spatial grid, term-by-term:

- **Log warping ($\psi$) is preserved per axis.** Both $\psi(|x_i - x_j|)$ and $\psi(|y_i - y_j|)$ use the same monotonic concave transform $\psi(x) = \log(1 + cx)$ as 1D FIRE, allocating higher attention resolution to small spatial offsets in each spatial direction with the same learnable scale $c$.

- **Progressive interpolation is preserved bi-axially.** Each per-axis term $\psi(d)/\psi(\max\{L, q\})$ is individually bounded in $[0, 1]$ for any spatial coordinate, so the MLP input stays in-distribution regardless of how $H_p$ or $W_p$ grow at inference. The bounded-input guarantee that underpins 1D FIRE's length-extrapolation behaviour therefore continues to hold *independently on both spatial axes*.

- **Query-position normalization is preserved per axis.** The $\max\{L, \cdot\}$ denominator carries over symbol-for-symbol — once for $x_i$ and once for $y_i$ — with no new quantities introduced. This is the structural element that no Euclidean 2D extension can preserve.

- **Per-head MLP structure is preserved.** The signature $f_\theta : \mathbb{R} \to \mathbb{R}^H$ is identical to 1D FIRE; only the scalar input to the MLP changes from a single normalized distance to the sum of two normalized distances.

- **2D spatial structure is preserved.** The quantities $|x_i - x_j|$ and $|y_i - y_j|$ are true grid distances computed from the actual patch coordinates, so the bias is a genuine function of 2D spatial offsets, not a flat sequence index. Two patches one row apart receive a meaningfully different bias from two patches one column apart, as required for image data.

- **Resolution-agnosticism is preserved.** The formula contains no fixed-size positional tables and depends only on on-the-fly grid coordinates, so the same parameters apply at any $H_p \times W_p$ patch grid without retraining or interpolation.

In a precise sense, axis decomposition is the 2D construction obtained by instantiating 1D FIRE once per spatial axis and summing the two normalized outputs — no new design element is introduced beyond the original formulation, and FIRE's $\max\{L, i\}$ normalizer is the *only* normalizer that appears anywhere in the 2D formula. To our knowledge this is the unique 2D extension with this property: the flattened and Euclidean variants each require modifications that have no 1D counterpart, while axis decomposition extends 1D FIRE symbol-for-symbol. The resulting bias depends only on relative offsets $(|x_i - x_j|, |y_i - y_j|)$ modulo the query-position normalizer that FIRE itself prescribes, so 2D translation invariance is preserved.

**Relationship to ALiBi, and family-level comparison fairness.** ALiBi extends to 2D using the Euclidean patch distance $d(i, j) = \|\mathbf{p}_i - \mathbf{p}_j\|_2$ (described in the next subsection). This is the natural extension of ALiBi because the 1D ALiBi bias is simply $-m_h \cdot |i - j|$, with no per-position normalizer to preserve; once a 2D distance metric is chosen, ALiBi can adopt it directly without introducing any new design element. FIRE has a per-position normalizer that ALiBi does not, so the natural 2D distance metric for FIRE and the natural 2D distance metric for ALiBi are *not* the same metric: the choice is dictated, in each case, by which 1D structural element the method must preserve under extrapolation. This metric asymmetry is intrinsic to the methods themselves, not an artifact of our experimental setup, and it would arise under any faithful 2D adaptation of the two methods.

Crucially, this does not compromise the family-level comparison that this paper is designed to make. Both FIRE-2D and ALiBi-2D operate on true 2D grid coordinates, both express the attention bias as a smooth function of genuine spatial offsets, and both are resolution-agnostic by construction. The rotation-based methods we compare against, RoPE and YaRN (described above and below respectively), likewise use true 2D phase rotations $e^{i(\theta_t^x p_n^x + \theta_t^y p_n^y)}$ computed from the same patch grid. The bias-vs.-rotation distinction we evaluate is therefore tested on a clean methodological footing: every method is evaluated in its most natural 2D realization, computed from the same underlying spatial coordinates, and no method is handicapped by a flattened, mismatched, or otherwise unnatural formulation. The metric choice *within* the additive-bias family (axis-decomposable for FIRE vs. Euclidean for ALiBi) is itself a design dimension that our experiments implicitly evaluate, and we discuss its empirical consequences — alongside the broader bias-vs.-rotation trends — in Sec. 5.

**Implementation details.** The same FIRE-2D module described by Eq. 10 is used unmodified across all three tasks — classification, detection, and segmentation — ensuring methodological consistency in our 2D evaluation. The learnable scalar $s$ (initialized to 0.1) acts as a global multiplier on the MLP output; it is not present in the original 1D FIRE, but in 2D the sum of two normalized terms widens the pre-MLP input range, and we found that an explicit learnable scale empirically stabilizes early optimization without altering the resolution-agnostic structure. The MLP $f_\theta$ is a two-layer network of hidden width 32 with GELU activations; $L$ is parameterized as $|L_{\text{init}} \cdot L_{\text{mult}}|$ with frozen $L_{\text{init}} = 14$ (matching the training patch-grid side length) and learnable multiplier $L_{\text{mult}}$; the warping scale $c$ inside $\psi$ is also learnable. For classification, the bias is padded so the class token receives zero positional offset; detection and segmentation backbones omit

the class token and apply the bias directly to patch tokens. The construction is continuous, contains no fixed-size positional tables, and naturally generalizes to larger patch grids at inference without interpolation or frequency rescaling.

### 3.4 YaRN: Yet another RoPE extension for Resolution Extrapolation

YaRN (Yet another RoPE eNhancement) Peng et al. (2024) is a frequency-aware extension of Rotary Position Embeddings designed to enable robust context-length extrapolation by selectively rescaling rotational frequencies and attention magnitudes. Unlike uniform interpolation of RoPE Chen et al. (2024), YaRN preserves high-frequency components responsible for local ordering while only interpolating low-frequency dimensions that encode global structure.

Recall that RoPE applies a complex rotation to query and key vectors:

$$\tilde{\mathbf{q}}_i = \mathbf{q}_i \odot e^{\mathrm{i}\theta(i)}, \qquad \tilde{\mathbf{k}}_j = \mathbf{k}_j \odot e^{\mathrm{i}\theta(j)}, \tag{12}$$

where $\theta_d(i) = i/\lambda_d$ and $\lambda_d$ denotes the wavelength of the $d$-th frequency. Position interpolation methods scale all dimensions uniformly, which compresses local relative angles and degrades short-range discrimination. YaRN instead performs *targeted interpolation* by modifying the RoPE frequencies as

$$\theta'_d = (1 - \gamma_d)\frac{\theta_d}{s} + \gamma_d \theta_d, \tag{13}$$

where $s$ is the extrapolation factor and $\gamma_d \in [0, 1]$ is a ramp function determined by the ratio between the original context length and the wavelength $\lambda_d$. Low-frequency dimensions (global structure) are interpolated, while high-frequency dimensions (local structure) are preserved.

In addition, YaRN introduces attention temperature scaling by rescaling the rotary embeddings, effectively modifying the attention logits as

$$\mathrm{Attn}(Q, K, V) = \mathrm{softmax}\left(\frac{QK^\top}{t\sqrt{d}}\right) V, \tag{14}$$

where $t$ is a scale-dependent temperature that stabilizes attention entropy under long-context extrapolation.

For Vision Transformers, we adapt YaRN to the 2D spatial domain by applying RoPE independently along horizontal and vertical axes. Let $(x_i, y_i)$ denote the coordinates of the $i$-th patch. The complex rotation is computed using mixed 2D frequencies:

$$\theta(i) = \omega_x x_i + \omega_y y_i, \tag{15}$$

with YaRN-scaled frequency matrices $\omega_x, \omega_y$ constructed per attention head and per layer. Crucially, in our experiments, YaRN is applied purely at inference time to a pre-trained RoPE model, without any additional fine-tuning. The modified frequencies are precomputed using the NTK-by-parts ramp and attention scaling, and the resulting complex exponentials are applied to queries and keys before attention computation.

This 2D YaRN formulation yields resolution-agnostic rotary embeddings that preserve local spatial precision while enabling stable zero-shot extrapolation to significantly larger patch grids than those observed during training, all while reusing the original RoPE weights.

### 3.5 ALiBi: Attention with Linear Biases for Vision Transformers

ALiBi Press et al. (2022) incorporates relative positional information into self-attention by adding a deterministic, head-specific linear bias directly to the attention logits. Unlike absolute positional embeddings, ALiBi does not encode positions explicitly, enabling robust extrapolation to longer sequences and higher resolutions without interpolation or additional learned positional parameters.

For a sequence indexed by positions $i$ and $j$, the attention bias for head $h$ is defined as

$$B_{ij}^{(h)} = -m_h \cdot d(i, j),$$

Table 1: Comparison of positional encoding methods. "Pos. Params" indicates learnable positional parameters; "Res. Agnostic" indicates whether the method handles arbitrary resolutions without interpolation.

| Method | Type | Pos. Params | Res. Agnostic |
|--------|------|-------------|---------------|
| Simple ViT | Absolute | $N \times d$ | No (interp.) |
| RoPE | Rotation | Freq. vectors | Partial |
| YaRN | Rotation | Same as RoPE | Yes (scaling) |
| ALiBi | Add. bias | 0 | Yes |
| FIRE | Add. bias | MLP ($\sim$1K) | Yes |

where $m_h$ is a fixed slope assigned to each head and $d(i, j)$ denotes the relative distance between tokens. The slopes decrease monotonically across heads, allowing different heads to focus on different context ranges. Since these slopes are fixed, ALiBi introduces no trainable parameters for positional encoding.

In vision transformers, tokens correspond to image patches arranged on a two-dimensional grid with spatial coordinates $(x_i, y_i)$. The relative distance between two patches is computed using the Euclidean metric

$$d(i, j) = \sqrt{(x_i - x_j)^2 + (y_i - y_j)^2},$$

preserving spatial inductive bias while remaining resolution-agnostic.

The classification token, which lacks a spatial location, is assigned a constant distance to all patch tokens, resulting in a uniform CLS-to-patch bias while maintaining meaningful patch-to-patch relationships. The resulting bias tensor is shared across all transformer layers.

During attention computation, the ALiBi bias is additively incorporated into the scaled dot-product attention,

$$\mathrm{Attn}(Q, K, V) = \mathrm{softmax}\left(\frac{QK^\top}{\sqrt{d_k}} + B\right)V,$$

directly modulating attention scores based on relative distance. The model otherwise follows a standard Vision Transformer architecture, with patch embeddings generated by a convolutional stem and final predictions obtained from the encoded CLS token.

### 3.6 Summary of Positional Encoding Methods

Table 1 summarizes the key properties of each positional encoding method. A fundamental distinction is between *additive bias* methods (ALiBi, FIRE), which inject positional information directly into attention logits, and *rotation-based* methods (RoPE, YaRN), which encode position through query-key rotations. Bias-based methods are inherently resolution-agnostic since they compute distances on-the-fly, while rotation-based methods require careful frequency design to avoid out-of-distribution phases at extrapolated resolutions.

## 4 Experiments

### 4.1 Experimental Setup

**Architecture.** Due to computational constraints, we conduct all experiments using a ViT-Tiny backbone with hidden dimension $d = 192$, depth $L = 12$, $H = 3$ attention heads, MLP dimension 768, and patch size $P = 16$. This yields approximately 5.7M parameters across all variants and ensures that any performance differences are attributable solely to the positional encoding mechanism. While larger models may exhibit different absolute performance, we expect the relative trends between positional encoding methods to generalize, as confirmed by the consistency of our findings across three diverse tasks.

**Training: Classification.** Models are trained from scratch on ImageNet-100 (a 100-class subset of ImageNet-1K Deng et al. (2009)) at resolution $224 \times 224$ for 300 epochs. We use AdamW Loshchilov

& Hutter (2019) with base learning rate $10^{-3}$, minimum learning rate $10^{-5}$, weight decay 0.05, and $(\beta_1, \beta_2) = (0.9, 0.999)$. The learning rate follows a cosine schedule with 10 epochs of linear warmup from $10^{-6}$. We apply label smoothing with factor 0.1, gradient clipping at norm 1.0, and automatic mixed precision. Data augmentation includes random resized cropping and horizontal flipping. For models with learnable frequency parameters (RoPE and YaRN), frequency tensors and bias/normalization parameters are excluded from weight decay.

**Training: Detection.** For object detection, we fine-tune on COCO 2017 Lin et al. (2014) using Faster R-CNN Ren et al. (2015) with an FPN Lin et al. (2017) neck built on top of each ViT-Tiny backbone. Models are initialized from the corresponding ImageNet-100 classification checkpoint. Training follows standard MMDetection Chen et al. (2019) protocols at a base resolution of $512 \times 512$.

**Training: Segmentation.** For semantic segmentation, we fine-tune on ADE20K Zhou et al. (2017) using UPerNet Xiao et al. (2018) with each ViT-Tiny backbone. Models are initialized from classification checkpoints and trained following standard MMSegmentation MMSegmentation Contributors (2020) protocols at a base resolution of $512 \times 512$.

**Positional Embedding Variants.** We compare the following methods:

- **Simple ViT**: Learned absolute 2D positional embeddings with bicubic interpolation at test time.

- **RoPE ViT**: 2D Rotary Position Embeddings with per-head random axis rotations ($\theta = 10$).

- **YaRN ViT**: Trained identically to RoPE (loaded from the same checkpoint); at inference, NTK-by-parts frequency scaling is applied dynamically based on the resolution ratio ($\theta = 10{,}000$, $\beta_{\text{fast}} = 32$, $\beta_{\text{slow}} = 1$).

- **ALiBi ViT**: Attention with Linear Biases using 2D Euclidean distance and fixed geometric head slopes. No learnable positional parameters.

- **FIRE ViT**: Continuous learnable attention bias parameterized by a small MLP with logarithmic distance warping and progressive interpolation.

**Evaluation Protocol.** For classification, we evaluate each model on the ImageNet-100 validation set at nine resolutions: 96, 160, 224 (train), 256, 384, 448, 512, 768, and 1024 pixels. This corresponds to scale factors ranging from $0.43\times$ (interpolation) to $4.57\times$ (extreme extrapolation) relative to the $14 \times 14$ training patch grid. All evaluations are performed *zero-shot* without any fine-tuning at the target resolution. For detection and segmentation, we test at resolutions 384, 512 (train), 640, 768, and 1024 pixels.

## 4.2 Classification Results

Table 2 reports Top-1 and Top-5 accuracy across resolutions. All models peak at or slightly above the training resolution, but diverge significantly under extrapolation.

Figure 1 visualizes these trends. While RoPE ViT attains the highest accuracy near the training resolution, FIRE ViT and ALiBi ViT remain substantially more stable under extreme resolution scaling.

**Key Observations. (i) Bias-based methods extrapolate best.** FIRE ViT achieves the highest accuracy at every resolution above 384, retaining 63.14% at 1024 (only 11.34 pp below its 224 baseline). ALiBi ViT exhibits a similarly graceful degradation profile, dropping just 12.42 pp from 224 to 1024. Both methods model relative distance through additive attention biases that are inherently resolution-agnostic.

**(ii) Raw RoPE collapses at high extrapolation.** RoPE ViT achieves the best accuracy at the training resolution (77.30%) but degrades catastrophically under extrapolation, falling to 18.90% at 1024, a 58.40 pp drop. This is consistent with the known sensitivity of RoPE to out-of-distribution positional frequencies.

Table 2: Zero-shot classification accuracy (%) on ImageNet-100 at varying resolutions. All models are trained at $224 \times 224$. Bold indicates best per resolution; underline indicates second best. YaRN uses the same checkpoint as RoPE but applies NTK-by-parts frequency scaling at inference.

| Model | Metric | Interpolation | | | Extrapolation | | | | | |
|---|---|---|---|---|---|---|---|---|---|---|
| | | 96 (0.43×) | 160 (0.71×) | 224 (1.0×) | 256 (1.14×) | 384 (1.71×) | 448 (2.0×) | 512 (2.29×) | 768 (3.43×) | 1024 (4.57×) |
| Simple ViT | Top-1 | 36.62 | 63.82 | 71.94 | 72.58 | 72.50 | 72.22 | 70.08 | 60.90 | 51.42 |
| | Top-5 | 60.54 | 84.54 | 89.94 | 90.60 | 90.34 | 90.18 | 89.72 | 84.98 | 78.40 |
| RoPE ViT | Top-1 | **45.46** | 70.00 | **77.30** | **78.08** | 76.46 | 72.10 | 65.90 | 36.42 | 18.90 |
| | Top-5 | **69.58** | 87.52 | 91.80 | 92.60 | 92.30 | 89.92 | 86.66 | 64.04 | 42.94 |
| YaRN ViT | Top-1 | 43.04 | **70.64** | 77.28 | 77.94 | 77.52 | 76.06 | 74.52 | 64.14 | 53.44 |
| | Top-5 | 68.00 | **88.08** | **91.80** | **92.64** | **93.34** | 92.64 | 91.64 | 86.66 | 79.12 |
| ALiBi ViT | Top-1 | 1.88 | 62.20 | 72.60 | 74.42 | 75.04 | 74.30 | 73.40 | 68.14 | 60.18 |
| | Top-5 | 7.86 | 83.62 | 89.48 | 90.48 | 91.30 | 91.30 | 90.60 | 88.10 | 84.16 |
| FIRE ViT | Top-1 | 42.28 | 65.90 | 74.48 | 76.00 | **77.88** | **77.40** | **77.22** | **71.06** | **63.14** |
| | Top-5 | 66.10 | 85.74 | 90.72 | 91.84 | 92.94 | **93.04** | **92.94** | **91.08** | **86.80** |

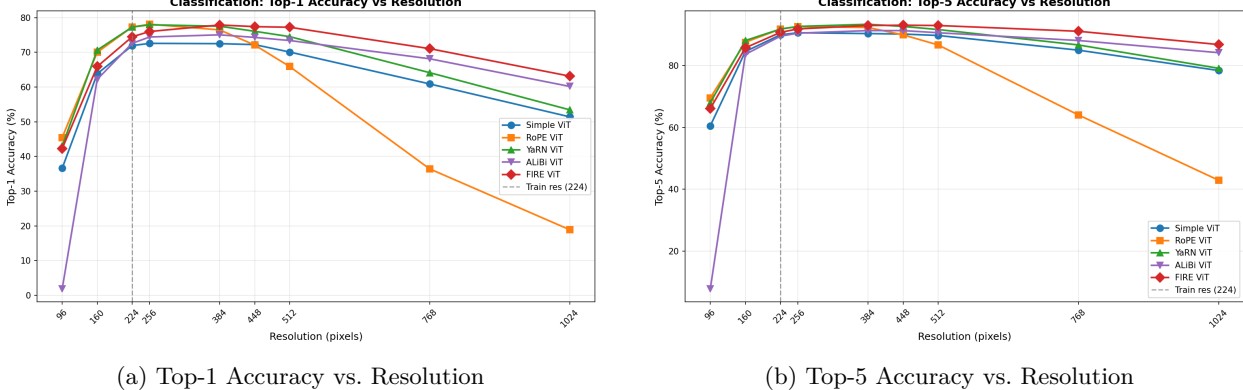

(a) Top-1 Accuracy vs. Resolution        (b) Top-5 Accuracy vs. Resolution

Figure 1: Zero-shot classification extrapolation on ImageNet-100. All models are trained at $224 \times 224$ (dashed line) and evaluated up to $1024 \times 1024$ without fine-tuning. FIRE ViT and ALiBi ViT exhibit the most robust extrapolation behavior, while raw RoPE ViT degrades sharply beyond the training resolution.

**(iii) YaRN significantly improves over raw RoPE.** YaRN's NTK-by-parts scaling yields substantial gains over raw RoPE across all extrapolation regimes. At moderate extrapolation (77.52% vs. 76.46% at 384), the improvement is modest, but at extreme scales the gap widens dramatically: YaRN retains 53.44% at 1024 compared to RoPE's 18.90%, a 34.54 pp advantage. This demonstrates that selective frequency scaling is highly effective for rotation-based methods, though YaRN still falls short of bias-based methods (FIRE: 63.14%, ALiBi: 60.18%) at the largest scale.

**(iv) Simple ViT is surprisingly competitive.** Bicubic interpolation of learned absolute embeddings provides a strong baseline, outperforming raw RoPE and YaRN at high extrapolation while remaining worse than ALiBi and FIRE.

### 4.3 Confirmation Experiment: Tiny-ImageNet Extrapolation

To confirm that our findings are not an artifact of the specific ImageNet-100 dataset or the $224 \times 224$ training resolution, we conduct an additional classification experiment on Tiny-ImageNet (200 classes). We train all models from scratch at a native resolution of $64 \times 64$ ($4 \times 4$ patches) for 100 epochs, and evaluate zero-shot extrapolation up to $512 \times 512$ ($8\times$ scale factor).

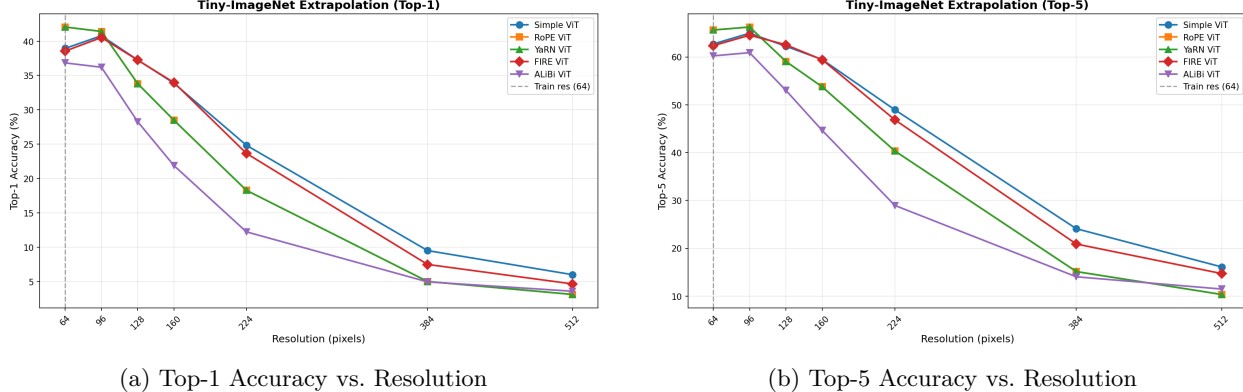

(a) Top-1 Accuracy vs. Resolution | (b) Top-5 Accuracy vs. Resolution

Figure 2: Zero-shot extrapolation on Tiny-ImageNet. Models trained at $64 \times 64$ and evaluated up to $512 \times 512$ ($8\times$ scaling). Among relative position encodings, FIRE maintains superior robustness compared to RoPE. YaRN fails to activate due to the small base resolution, and Simple ViT acts as a surprisingly strong baseline.

Table 3: Detection performance (%) on COCO at varying resolutions. Trained at $512 \times 512$. $AP_{50}$ provides a more interpretable view of the extrapolation trends.

| Model | Metric | 384 | 512 | 640 | 768 | 1024 |
|---|---|---|---|---|---|---|
| Simple ViT | mAP | 3.7 | 5.6 | 3.5 | 3.3 | 2.5 |
| | $AP_{50}$ | 8.4 | 12.8 | 8.0 | 7.3 | 5.5 |
| RoPE ViT | mAP | 6.0 | 6.8 | 6.0 | 4.5 | 1.6 |
| | $AP_{50}$ | 13.2 | 15.1 | 13.7 | 10.4 | 3.7 |
| YaRN ViT | mAP | 6.0 | 6.8 | 6.1 | 4.8 | 2.7 |
| | $AP_{50}$ | 13.0 | 15.1 | 13.9 | 11.2 | 6.2 |
| ALiBi ViT | mAP | 3.0 | 3.4 | 3.4 | 3.3 | 2.7 |
| | $AP_{50}$ | 6.6 | 7.6 | 7.6 | 7.2 | 5.9 |
| FIRE ViT | mAP | 3.4 | 3.4 | 3.2 | 2.9 | 2.3 |
| | $AP_{50}$ | 7.4 | 7.5 | 6.8 | 6.1 | 4.7 |

As shown in Figure 2, training on extremely small grids introduces unique dynamics, but the relative advantage of bias-based methods over rotation-based methods holds. At a $3.5\times$ scale factor (224 px), FIRE ViT retains 23.66% Top-1 accuracy, substantially outperforming RoPE (18.28%). At extreme $8\times$ scaling (512 px), FIRE continues to lead the relative methods (4.64% vs. RoPE's 3.11%). Interestingly, Simple ViT provides the strongest overall performance (24.84% at 224 px, 5.99% at 512 px); bicubic interpolation of a $4 \times 4$ grid yields an exceptionally smooth spatial prior that works well, whereas relative methods struggle because they only observed distance offsets up to 3 pixels during training.

Furthermore, this experiment reveals a fundamental limitation of YaRN on small base resolutions. YaRN performs *identically* to raw RoPE across all extrapolated resolutions. Because the training grid is only $4 \times 4$, the maximum context length is too small to trigger YaRN's frequency scaling formula (the interpolation thresholds evaluate to negative values). This means YaRN requires a sufficiently large training resolution to engage its extrapolation mechanisms, unlike FIRE which dynamically adapts.

## 4.4 Detection Results

Table 3 reports zero-shot bbox mAP on COCO under resolution extrapolation. Detection models are trained at $512 \times 512$ and evaluated at resolutions up to 1024.

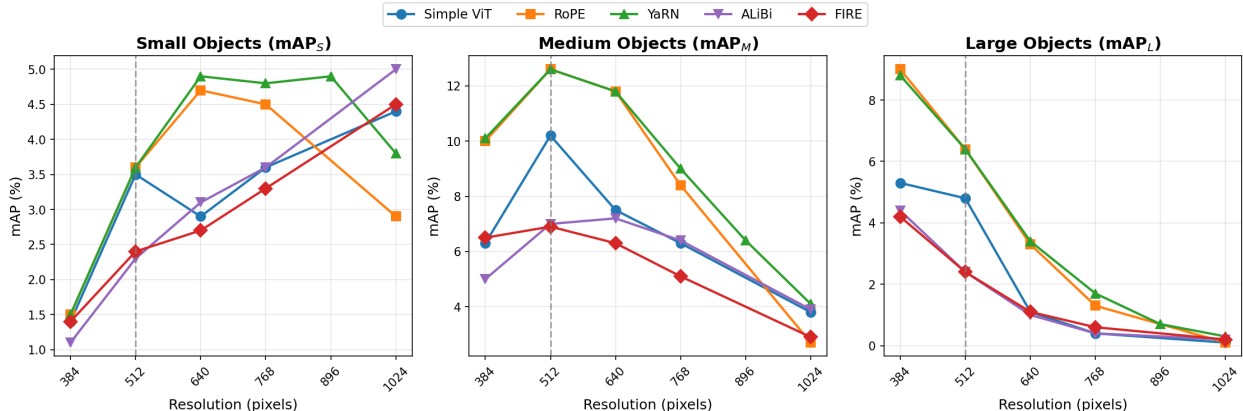

Figure 3: Detection mAP breakdown by object size across resolutions. Large objects ($\text{mAP}_L$) degrade most rapidly under extrapolation, while small object detection ($\text{mAP}_S$) improves at higher resolutions for bias-based methods.

Table 4: Segmentation mIoU (%) on ADE20K at varying resolutions. Trained at $512 \times 512$.

| Model | 384 | 512 | 640 | 768 | 1024 |
|---|---|---|---|---|---|
| Simple ViT | 16.5 | 21.8 | 16.0 | 14.5 | 11.5 |
| RoPE ViT | 24.5 | **25.0** | 23.1 | 19.3 | 9.2 |
| YaRN ViT | **24.6** | 24.9 | 23.7 | 20.9 | 15.4 |
| ALiBi ViT | 22.2 | 23.5 | 23.6 | **23.0** | **20.5** |
| FIRE ViT | 23.4 | 24.4 | **24.2** | 22.5 | 19.8 |

In detection, RoPE ViT achieves the highest mAP at the training resolution (6.8%) but experiences the steepest decline, dropping to 1.6% at 1024, a 75% relative drop in $\text{AP}_{50}$ (from 15.1% to 3.7%). ALiBi ViT demonstrates the most stable extrapolation profile, with only a 22% relative drop in $\text{AP}_{50}$ from 512 to 1024. Notably, the overall mAP values are low due to the limited capacity of ViT-Tiny and training on ImageNet-100 rather than full ImageNet-1K; however, the *relative* degradation patterns are consistent with the classification findings and confirm that positional encoding robustness transfers to dense prediction.

**Object Size Breakdown.** Figure 3 decomposes detection performance by object scale (small, medium, large). A striking finding is that *large objects degrade most rapidly* under extrapolation across all methods. At 1024 px, $\text{mAP}_L$ collapses to near-zero for most methods ($< 0.01$), while $\text{mAP}_S$ for small objects actually *increases* for bias-based methods: ALiBi reaches 5.0% and FIRE reaches 4.5%. This counterintuitive result suggests that higher resolution provides more pixels per small object, improving detection, while large objects suffer from attention pattern fragmentation. RoPE exhibits the most severe large-object collapse ($9.0\% \rightarrow 0.1\%$), whereas ALiBi maintains the most balanced performance across scales.

## 4.5 Segmentation Results

Table 4 reports mIoU on ADE20K under resolution extrapolation.

Segmentation results reinforce the classification trends. ALiBi ViT retains 20.5% mIoU at 1024 (only 3.0 pp below its peak), while RoPE ViT degrades from 25.0% to 9.2%, a 15.8 pp collapse. FIRE ViT similarly maintains strong performance, dropping only 4.6 pp from peak to 1024. Simple ViT suffers a 10.3 pp decline, confirming that interpolated absolute embeddings provide weaker spatial coherence than relative or bias-based alternatives for dense prediction at novel resolutions.

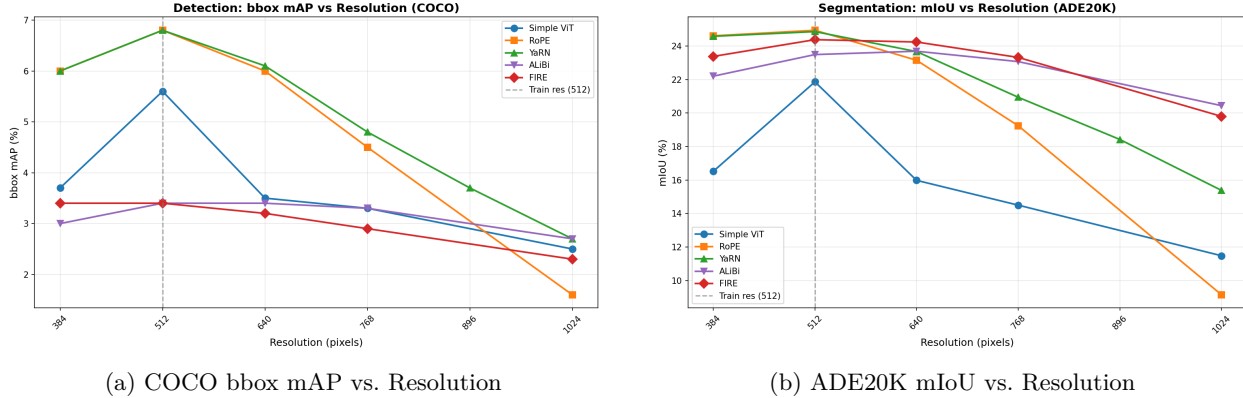

(a) COCO bbox mAP vs. Resolution        (b) ADE20K mIoU vs. Resolution

Figure 4: Zero-shot extrapolation on dense prediction tasks. All models are trained at $512 \times 512$ (dashed line) and evaluated up to $1024 \times 1024$ without fine-tuning. ALiBi ViT and FIRE ViT maintain stable performance at higher resolutions, while RoPE ViT degrades sharply. YaRN provides moderate improvement over RoPE but still degrades at extreme scales.

Figure 4 shows that the extrapolation trends observed in classification transfer to dense prediction tasks as well. Relative-bias methods remain substantially more stable than rotation-based methods as evaluation resolution increases.

### 4.6 Analysis

**Bias vs. Rotation Methods.** Across all three tasks, the additive bias methods (ALiBi and FIRE) consistently outperform the multiplicative rotation methods (RoPE and YaRN) under resolution extrapolation. We hypothesize that this is because additive biases directly modulate attention logits as a smooth function of distance, making them naturally resolution-agnostic. In contrast, rotation-based methods encode position through phase angles that grow linearly with coordinate magnitude; when extrapolated beyond the training range, these phases become out-of-distribution, causing destructive interference in the attention pattern.

**The Role of Frequency Awareness.** Among rotation-based methods, YaRN improves over raw RoPE through selective frequency scaling: it preserves high-frequency (local) components while interpolating low-frequency (global) ones via the NTK-by-parts ramp. While this yields gains at moderate extrapolation, YaRN still degrades substantially at extreme scales ($4.57\times$), suggesting that the ramp parameters ($\beta_{\text{fast}}$, $\beta_{\text{slow}}$) may require resolution-specific tuning for 2D spatial data.

**Consistency Across Tasks.** The extrapolation rankings observed in classification (FIRE > ALiBi > Simple > YaRN > RoPE) are largely preserved in detection and segmentation, indicating that positional encoding robustness is a fundamental architectural property rather than a task-specific artifact. This consistency provides practical guidance: if a model extrapolates well in classification, it will likely do so in dense prediction as well.

## 5 Comparison and Analysis

This section synthesizes the empirical trends observed in Section 4 and provides a comparative interpretation of how different positional encoding mechanisms behave under resolution extrapolation. We focus on four recurring themes: (i) the distinct extrapolation profiles visible in classification, (ii) the structural difference between additive bias methods and rotation-based methods, (iii) the contrasting behavior of inference-time frequency scaling strategies, and (iv) the transfer of these trends to dense prediction tasks.

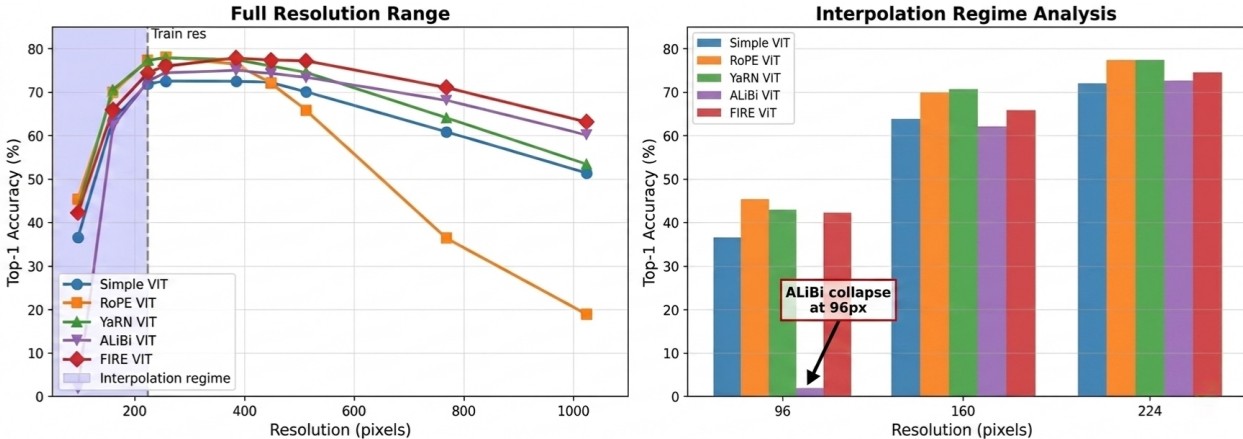

Figure 5: Interpolation regime analysis. Left: full resolution sweep with interpolation regime highlighted. Right: bar chart of sub-training resolutions showing ALiBi's anomalous collapse at 96 px (1.88%) while other methods retain 36–45% accuracy.

## 5.1 Classification Extrapolation Trends

The classification results reveal three qualitatively distinct operating regimes as the input resolution moves away from the training point. In the sub-training regime (96–160 px), all methods experience some degradation due to reduced spatial detail, although the extent varies substantially. In the near-training regime (224–384 px), several methods improve beyond their training-resolution accuracy, indicating that higher-resolution inputs can provide genuinely useful additional spatial information when the positional encoding remains well behaved. In the far-extrapolation regime (448–1024 px), the choice of positional encoding becomes the dominant factor, and the performance gap between methods widens sharply.

**Interpolation Regime Anomaly: A Deep Ablation.** Figure 5 exposes a severe vulnerability in the interpolation regime: ALiBi ViT collapses to 1.88% Top-1 accuracy at 96 px, a catastrophic degradation from its 74.40% training-resolution performance. In contrast, all other methods retain robust accuracies between 36% and 45%.

To isolate the mechanism behind this asymmetric failure, we reproduced the phenomenon using a Tiny-ImageNet model trained at 64 px and evaluated at 32 px. At 32 px, Simple ViT degrades conventionally from 36.8% to 10.0%. ALiBi, however, collapses entirely to 1.15%. We tested three hypotheses for this failure: (A) the fixed CLS token distance penalty disproportionately dominates attention on smaller grids; (B) the 50% reduction in maximum physical patch distance eliminates the large negative penalties required to suppress background tokens; and (C) a combination of both effects.

Explicit recalibration of these parameters during inference conclusively falsified all three hypotheses. Sweeping the CLS token distance from 0.0 to 10.0 yielded a peak recovery of only 1.71%. Scaling inter-patch slope multipliers by 1.5× to 3.0× to restore the training distance range peaked at 1.12%. Jointly scaling both parameters yielded 0.81%.

These negative results prove that ALiBi's interpolation collapse stems from a fundamental topological flaw rather than a parameter calibration issue. Because ALiBi employs unnormalized, static relative distances without absolute positional anchors, it depends entirely on local structural neighborhoods. On a $14 \times 14$ training grid, the model learns distinct attention patterns for fully surrounded center tokens versus edge tokens. When interpolated to a $6 \times 6$ or $2 \times 2$ grid, the proportion of edge tokens spikes, and the expected center-token topology is destroyed. Unlike FIRE, which normalizes spatial distances to preserve topological relationships across scales, ALiBi's static slopes cannot adapt to this structural shift, causing the learned token interactions to fail completely.

At the largest evaluated scale (1024 px, corresponding to 4.57× the training grid), the Top-1 ranking is: FIRE (63.14%) > ALiBi (60.18%) > YaRN (53.44%) > Simple ViT (51.42%) > RoPE (18.90%). The gap between the best and worst methods at this scale is 44.24 percentage points, which is substantially larger than the 5.36-point spread at the training resolution (71.94%–77.30%). This demonstrates that extrapolation robustness, rather than in-distribution accuracy alone, is the primary differentiator among positional encoding strategies in resolution-flexible Vision Transformers.

The Top-5 curves reinforce the same ordering while showing a slightly more compressed spread. At 1024 px, FIRE achieves the highest Top-5 accuracy (86.80%), followed by ALiBi (84.16%) and YaRN (79.12%), while vanilla RoPE drops to 42.94%. This indicates that the strongest methods degrade gracefully: even when the top prediction changes, the correct class often remains within the top few candidates, suggesting that the underlying representation remains semantically coherent over a broad range of resolutions. Notably, YaRN's NTK-by-parts scaling substantially improves its Top-5 retention compared to raw RoPE, nearly doubling it at 1024 px.

### 5.2 Bias-Based vs. Rotation-Based Methods

A consistent pattern across the experiments is the stronger extrapolation behavior of additive bias-based methods (ALiBi and FIRE) compared to rotation-based methods (RoPE and YaRN). The key architectural distinction is that bias-based methods inject positional information as an additive term in the attention logits, whereas rotation-based methods modify the query and key representations themselves before the dot product is computed.

This difference has important consequences under extrapolation. In bias-based methods, positional information acts as a spatial prior that encourages locality or relative structure, but the underlying content-based similarity remains intact. As the resolution increases, the positional bias simply extends to larger spatial distances, which generally leads to gradual degradation rather than abrupt failure. This behavior is clearly visible in both ALiBi and FIRE, which remain strong even at 1024 px. FIRE is the most robust overall in classification, dropping only 11.34 percentage points from its 224 px Top-1 accuracy (74.48%) to 1024 px (63.14%), while ALiBi drops only 12.42 points over the same range.

In contrast, rotation-based methods encode position directly into the geometry of the query–key interaction. This yields strong in-distribution performance: RoPE achieves the highest Top-1 accuracy at the training resolution (77.30%) and the peak score at 256 px (78.08%), but becomes increasingly fragile when evaluated outside the training range. As the patch grid grows, the effective positional phases move beyond the range seen during training, and the resulting attention patterns become progressively less stable. This is most visible in vanilla RoPE, which falls from 77.30% at 224 px to 18.90% at 1024 px, a 58.40-point collapse.

A notable secondary result is the strength of the Simple ViT baseline. Despite relying on learned absolute embeddings with bicubic interpolation, it reaches 51.42% Top-1 at 1024 px, remaining competitive with YaRN (53.44%) under extreme extrapolation. This suggests that smooth interpolation of learned absolute embeddings provides a surprisingly strong "safety floor" when more sophisticated relative schemes become unstable outside their training range. However, YaRN's selective frequency scaling enables it to outperform Simple ViT by maintaining better local spatial precision while still benefiting from interpolated global structure.

### 5.3 Attention Pattern Analysis

To investigate *why* positional encodings differ so dramatically under resolution extrapolation, we examine attention patterns from the final transformer block. We report two complementary views: a qualitative visualization of CLS-token attention on a representative image (Fig. 7), and quantitative concentration metrics aggregated over 128 ImageNet-100 validation images at every test resolution (Fig. 6).

**A resolution-comparable concentration metric.** The standard measurement of attention diffusion, the summed entropy $H_{\text{total}} = -\sum_{ij} A_{ij} \log A_{ij}$, grows mechanically with the token count $N$. From 224 px ($N = 197$) to 1024 px ($N = 4097$), this metric inflates by roughly 20× independent of actual attention

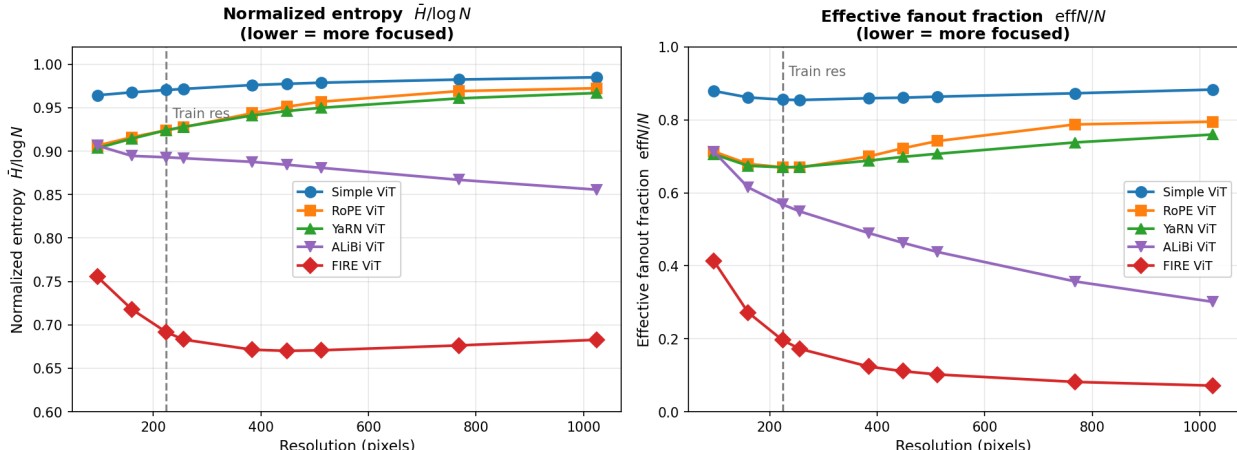

Figure 6: Resolution-comparable attention concentration metrics, computed on 128 ImageNet-100 validation images at the final encoder block of each model. *Left:* mean normalized per-query entropy $\bar{H}/\log N$ (lower = more focused). *Right:* effective fanout fraction $\mathrm{eff}N/N = \exp(\bar{H})/N$ (lower = more focused). Both metrics are intrinsically scale-free, so changes across resolution reflect changes in attention concentration rather than token-count inflation. Bias-based methods (ALiBi, FIRE) actively concentrate attention as resolution grows; rotation-based methods (RoPE, YaRN) and Simple ViT do not.

behavior, confounding cross-resolution comparisons. To isolate genuine attention concentration, we evaluate two scale-free metrics:

- **Mean normalized entropy** ($\bar{H}/\log N \in [0, 1]$): The average per-query entropy divided by its theoretical maximum. A value of 0 indicates perfectly peaked attention; 1 indicates uniform attention.

- **Effective fanout fraction** ($\mathrm{eff}N/N = \exp(\bar{H})/N \in (0, 1]$): The perplexity of the attention distribution divided by $N$. This quantifies the effective fraction of available tokens to which a query meaningfully attends.

Figure 6 reports these metrics for the final encoder block across nine test resolutions.

**Bias-based methods actively concentrate attention as resolution grows.** This normalized analysis isolates a stark mechanistic difference: additive bias methods progressively focus attention as resolution increases, whereas rotation-based methods distribute it diffusely. FIRE's effective fanout fraction drops monotonically from 0.20 at 224 px to 0.07 at 1024 px. At the largest scale, FIRE effectively attends to only 7% of the available tokens. ALiBi exhibits a similarly strong concentration trend, dropping from 0.57 at 224 px to 0.30 at 1024 px.

In sharp contrast, rotation-based methods fail to concentrate. RoPE's effective fanout fraction increases from 0.67 at 224 px to 0.79 at 1024 px. YaRN's selective frequency scaling provides marginal improvement but follows the same diffuse trajectory (0.67 → 0.76). Simple ViT exhibits the highest diffusion, maintaining a nearly uniform 0.85 to 0.88 fanout fraction across all scales. The normalized entropy metric confirms this identical hierarchy: FIRE maintains the lowest normalized entropy ($\sim 0.67$ to $0.76$), ALiBi decreases from 0.91 to 0.86, while RoPE and YaRN gradually inflate from 0.91 to 0.97.

**Reinterpretation of the mechanistic story.** These scale-free metrics reframe the fundamental distinction between bias-based and rotation-based architectures. Extrapolation failure is not characterized by a discrete collapse to uniformity, but by the absence of an active concentration mechanism. Bias-based methods inherently possess this mechanism: ALiBi's monotonically decaying $-m_h\, d(i, j)$ bias and FIRE's learned per-axis bias physically suppress contributions from spatially distant tokens. As absolute grid distances grow, this suppression strengthens, enforcing strict local attention. Rotation-based methods lack any

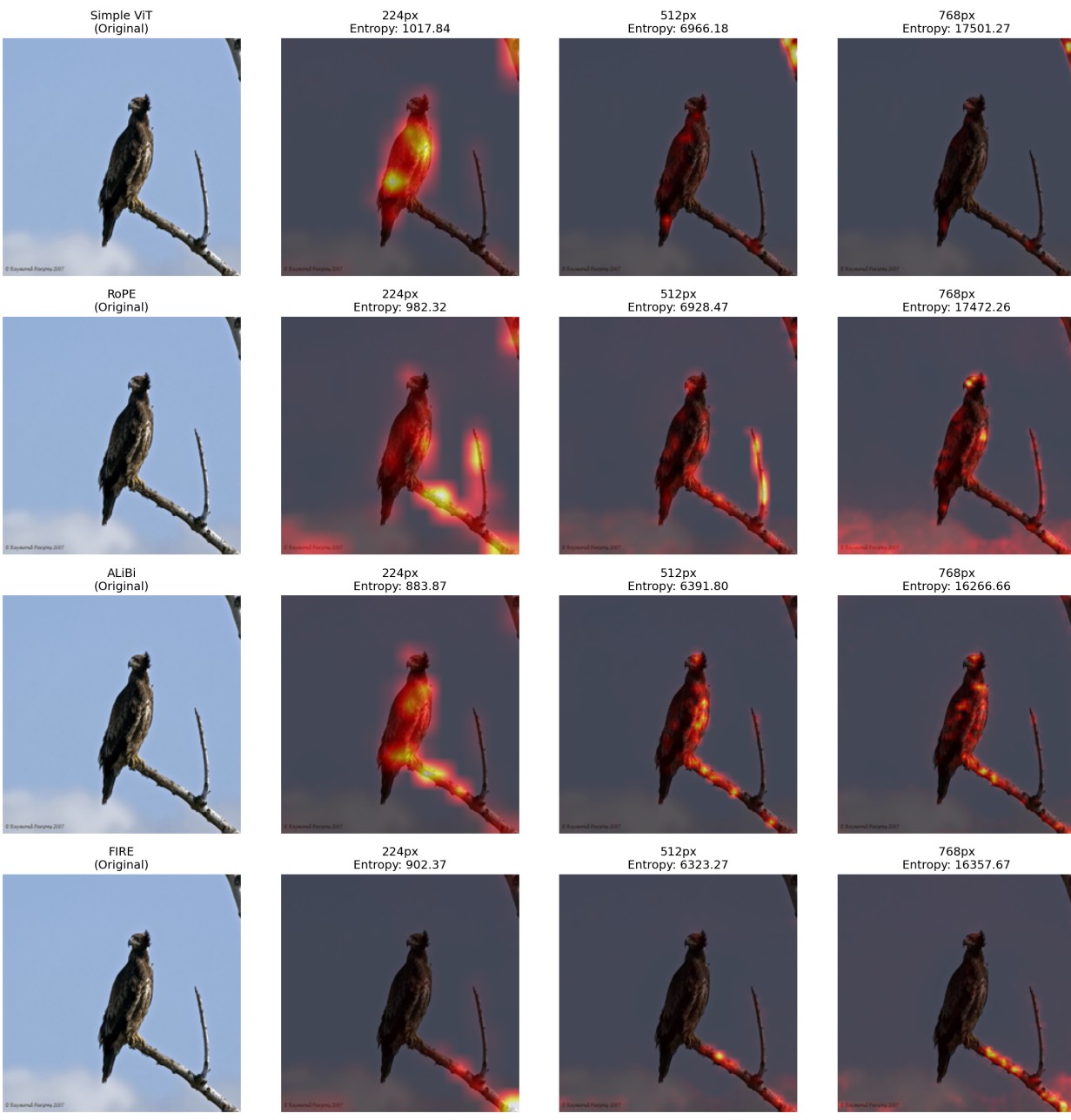

Figure 7: Qualitative CLS-attention visualization across resolutions on a representative image. Each row is a positional encoding method; columns show the original image and attention heatmaps at 224, 512, and 768 px. The entropy values overlaid on each panel are raw summed entropies $H_{\text{total}}$ and are reported here for visual cross-reference only; the resolution-comparable quantitative claims in the paper are based on the normalized metrics in Fig. 6. Visually, ALiBi and FIRE preserve concentration on the subject as resolution grows, while RoPE and Simple ViT spread attention across uninformative background regions.

comparable mechanism to attenuate out-of-distribution phase angles, forcing attention to spread uniformly across the expanded grid. The accuracy trends in Section 4 directly validate this dynamic: methods that maintain spatial focus (FIRE, ALiBi) retain classification accuracy at extreme scales, while methods that diffuse attention (RoPE) suffer catastrophic performance drops.

## 5.4 Dense Prediction and Practical Implications

The dense prediction experiments broadly support the trends observed in classification, but they also reveal an important nuance: the absolute ranking is not identical across all tasks, particularly in detection. In object detection, RoPE achieves the highest mAP near the training regime, reaching 6.8% at 512 px and remaining strong at 640 px (6.0%), but it also exhibits the steepest degradation, falling to 1.5% at 1024 px. YaRN, applying the same checkpoint with NTK-by-parts scaling, matches RoPE at 512 px (6.8%) but degrades more gracefully to 2.7% at 1024 px, nearly double RoPE's retained performance. By contrast, ALiBi shows the most stable detection profile, decreasing only from 3.4% at 512 px to 2.7% at 1024 px. FIRE remains competitive and relatively stable, dropping from 3.4% to 2.3%, but under the current ViT-Tiny setting it does not surpass RoPE in peak detection mAP. We note that the absolute COCO mAP values across all methods remain very low (e.g., peak of 6.8%). This is a direct consequence of using a lightweight ViT-Tiny backbone without large-scale pre-training or specialized detection necks. Consequently, these object detection results serve primarily to illustrate relative degradation trends under resolution shift, rather than to establish state-of-the-art absolute detection performance.

In semantic segmentation, the extrapolation pattern aligns more closely with the classification results. RoPE achieves the highest score at the training resolution (25.0% mIoU at 512 px), but degrades sharply to 9.2% at 1024 px. YaRN significantly mitigates this degradation, retaining 15.4% mIoU at 1024 px compared to RoPE's 9.2%, a 6.2 pp improvement from the same checkpoint. By contrast, ALiBi retains 20.5% at 1024 px, and FIRE retains 19.8%, indicating substantially stronger spatial robustness at large resolutions. FIRE also exhibits the strongest peak among the stable methods, reaching 24.4% at 512 px and remaining above 24% at 640 px, while ALiBi maintains a flatter and more consistent curve across the full resolution sweep. YaRN occupies a middle ground, preserving more spatial coherence than raw RoPE while falling short of the bias-based methods.

These results suggest that positional encoding failures are amplified in dense prediction, where the model must maintain spatially coherent attention patterns across many local regions rather than produce a single global classification decision. Classification can partially average out local errors through the final global representation, whereas detection and segmentation depend more directly on stable token-to-token spatial relationships. From a deployment perspective, this makes bias-based methods particularly attractive for multi-scale vision systems: they may sacrifice some peak in-distribution accuracy, but they provide significantly more reliable behavior when the input resolution shifts at test time.

Overall, our findings support a clear practical recommendation. If the target application requires robustness across varying or unseen resolutions, additive bias-based positional encodings, especially FIRE and ALiBi, offer the strongest trade-off between accuracy and stability. If maximizing in-distribution accuracy is the primary objective and the evaluation resolution closely matches the training setup, RoPE remains highly competitive, but its extrapolation fragility should be carefully considered. YaRN provides a practical middle ground: by selectively rescaling RoPE frequencies at inference time without retraining, it substantially improves extrapolation robustness over raw RoPE while retaining much of RoPE's in-distribution strength.

## 5.5 Computational Cost

Table 5 reports peak GPU memory and inference throughput across resolutions. All methods have similar parameter counts ($\sim$5.5M), but differ substantially in memory and compute overhead. Bias-based methods (ALiBi, FIRE) require storing the full $N \times N$ attention bias matrix, leading to higher memory at large resolutions: FIRE uses 6.8 GB at 1024 px versus 2.8 GB for Simple ViT. YaRN shows anomalously low memory due to implementation differences in frequency caching. Throughput follows inverse trends: Simple ViT achieves 113 img/s at 224 px while FIRE drops to 5.8 img/s at 1024 px.

These computational trade-offs should inform deployment decisions. While bias-based methods offer superior extrapolation, their 2–2.5$\times$ memory overhead at high resolutions may be prohibitive in resource-constrained settings. Conversely, if memory is limited but extrapolation robustness is required, YaRN provides a practical compromise with minimal overhead over RoPE.

Table 5: Computational cost at varying resolutions (batch size 1). Memory in MB; throughput in images/second. Bold indicates best (lowest memory / highest throughput); underline indicates second best.

| Model | Metric | 224 | 384 | 512 | 768 | 1024 |
|-------|--------|-----|-----|-----|-----|------|
| Simple ViT | Mem. | 38 | 90 | 207 | 901 | 2755 |
|            | Thr.  | **113** | **106** | **106** | **46** | **17** |
| RoPE ViT | Mem. | 40 | 96 | 216 | 921 | 2791 |
|          | Thr.  | 80 | 83 | 80 | 36 | 13 |
| YaRN ViT | Mem. | **35** | **48** | **73** | **195** | **491** |
|          | Thr.  | 75 | 64 | 67 | 33 | 13 |
| ALiBi ViT | Mem. | 44 | 141 | 362 | 1690 | 5249 |
|           | Thr.  | 99 | 101 | 94 | 28 | 10 |
| FIRE ViT | Mem. | 45 | 165 | 453 | 2165 | 6765 |
|          | Thr.  | 107 | 105 | 84 | 17 | 6 |

## 6 Conclusion

We presented a systematic study of positional encoding strategies for Vision Transformers under resolution extrapolation, evaluating learned absolute embeddings Dosovitskiy et al. (2021), RoPE Su et al. (2024); Heo et al. (2024), ALiBi Press et al. (2022), YaRN Peng et al. (2024), and FIRE Li et al. (2024) across classification, detection, and segmentation. Our experiments reveal a clear dichotomy: attention bias-based methods (ALiBi, FIRE) substantially outperform rotation-based methods (RoPE, YaRN) under extrapolation, with FIRE retaining 63.1% Top-1 accuracy at $4.6\times$ resolution compared to RoPE's 18.9%. A resolution-comparable attention concentration analysis traces this performance gap to a concentration mechanism: bias-based methods progressively focus attention on a small subset of tokens as resolution grows, while rotation-based methods leave attention broadly diffuse at all extrapolated resolutions.

**Practical Recommendations.** For applications requiring resolution flexibility, we recommend FIRE or ALiBi as they provide the best extrapolation-accuracy trade-off. RoPE remains competitive for fixed-resolution deployment where in-distribution accuracy is prioritized. YaRN offers a practical middle ground when modifying a pre-trained RoPE model, providing substantial extrapolation gains through inference-time frequency scaling without retraining.

**Limitations.** Due to computational constraints, our study focuses on ViT-Tiny ($\sim$5.7M parameters) trained on ImageNet-100. While the consistent trends across three diverse tasks suggest our findings generalize, validation on larger models (ViT-Base/Large) and full ImageNet-1K would strengthen these conclusions. We report single-seed results; however, the convergent patterns across classification, detection, and segmentation provide implicit robustness evidence. Additionally, the low absolute mAP values in detection reflect ViT-Tiny's limited capacity rather than fundamental limitations of the positional encoding methods.

**Future Directions.** Several avenues merit further investigation: (i) scaling analysis to determine whether extrapolation trends hold for larger ViT variants; (ii) learned frequency scaling for RoPE that adapts to 2D spatial structure rather than using fixed NTK-by-parts ramps; (iii) hybrid approaches combining the in-distribution strength of rotation-based methods with the extrapolation stability of bias-based methods; and (iv) extension to video transformers where temporal extrapolation introduces additional challenges. We hope this work provides a foundation for building resolution-robust vision architectures under practical compute constraints.

**Broader Impact.** Resolution-flexible Vision Transformers can reduce computational costs and carbon footprint by enabling training at lower resolutions while maintaining performance at deployment. This work contributes to more efficient vision systems, though practitioners should validate extrapolation behavior on their specific domains before deployment.

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
