# OpenReview forum: "Analysing Extrapolation Capabilities of Modern Positional Embeddings in Vision Transformers"
_TMLR — Under review for TMLR_

### Review · Reviewer_FPwW · 2026-06-09

**Summary Of Contributions:**

This paper presents an empirical study of zero-shot resolution extrapolation in Vision Transformers, focusing on how positional encoding affects performance when a model is trained at lower resolution and evaluated at substantially higher resolution without fine-tuning. The authors compare five positional encoding strategies including Simple ViT, RoPE, YaRN, ALiBi, and FIRE using a common ViT-Tiny backbone. The experiments include image classification training on ImageNet-100 at 224×224 and evaluation up to 1024×1024, plus downstream experiments on COCO detection and ADE20K segmentation with training at 512×512 and evaluation up to 1024.

The main reported finding is that additive attention-bias methods are more robust than rotation-based methods under large extrapolation factors. In the classification setting at 1024×1024, FIRE achieves 63.14% Top-1 accuracy, ALiBi 60.18%, YaRN 53.44%, Simple ViT 51.42%, and RoPE 18.90%, while RoPE is strongest near the training resolution. The paper also includes an analysis section arguing that bias-based methods preserve more stable attention patterns under extrapolation, and highlights an interesting failure mode of ALiBi at very low resolution, where performance collapses at 96 px.

**Additional Comments:**

I found the paper easy to read and generally well organized. The core classification result is interesting and potentially useful, especially the separation between methods that perform best near the training resolution and methods that remain stable under extreme extrapolation. My recommendation is mainly limited by the mismatch in 2D method adaptation, the lack of variance reporting, and the current strength of the mechanistic analysis.

**Audience:**

Yes

**Audience Explanation:**

The topic is relevant to researchers working on transformers, positional encodings, robustness under distribution shift, and efficient deployment of vision models. The paper studies a practically important question "whether models trained at lower resolution can be deployed at higher resolution without fine-tuning" and identifies a useful distinction between near-training accuracy and large-scale extrapolation robustness.

The paper would also interest readers who value careful empirical comparisons, because it places several widely used positional encoding strategies under a common evaluation protocol and extends the analysis across classification, detection, and segmentation.

**Broader Impact Concerns:**

I do not have major broader-impact concerns that would require additional discussion beyond what is already stated in the paper.

**Claims And Evidence:**

No

**Claims Explanation:**

The paper does provide clear evidence for a narrower claim: in the reported ViT-Tiny/ImageNet-100 setting, additive bias methods such as FIRE and ALiBi are substantially more robust than raw RoPE under large resolution extrapolation, and the segmentation results broadly reinforce that pattern. However, I do not think the current evidence fully supports some of the paper’s broader family-level and mechanistic conclusions.

My main concern is methodological comparability across the 2D adaptations. ALiBi is implemented using explicit 2D Euclidean distances on the image grid, whereas FIRE is described as operating on the flattened patch sequence using $|i-j|$. This matters because flattening a 2D grid into a 1D sequence changes spatial geometry and locality, so the central comparison between bias-based and rotation-based families is not as clean as the paper suggests.

A second concern is the attention-entropy analysis. Since the number of tokens changes substantially with resolution, entropy-based comparisons across resolutions need careful normalization, and aggregate entropy can increase mechanically with sequence length even without a meaningful change in attention quality. In its current form, the analysis is suggestive, but I do not think it is yet sufficient to strongly support the causal interpretation that rotation-based methods fail specifically because of the attention-collapse mechanism claimed in the paper.

A third concern is statistical rigor. The paper reports single tables and curves but does not provide multi-seed means or variance estimates, which makes it difficult to know how robust some of the reported rankings are, especially at moderate extrapolation scales and in downstream settings.

Finally, the dense-prediction evidence should be interpreted cautiously. The COCO mAP values are very low overall, with the best training-resolution mAP reported as 6.8, so while the detection experiments are still useful as trend evidence, they do not strongly support broad practical conclusions about object detection performance.

**Requested Changes:**

**Critical**
- Please make the 2D comparison fairer or narrow the conclusions accordingly. In particular, the paper should either evaluate a genuinely 2D version of FIRE or explain much more carefully why flattened $|i-j|$ is the appropriate 2D adaptation, since ALiBi uses explicit 2D Euclidean distance and the current comparison is not fully matched.

- Please revise the attention analysis using a resolution-comparable metric. At minimum, the entropy analysis should be normalized appropriately and reported per query, head, or layer, or supplemented with alternative concentration statistics, so that cross-resolution comparisons are not dominated by changing token count.

- Please add multi-seed variance reporting for the main classification results. Even if full multi-seed repetition is too expensive for all downstream tasks, at least the main classification table should report mean and standard deviation over multiple runs.

- Please align the scope of the claims with the evidence provided. In particular, the authors should either add one additional confirmation experiment beyond the current ViT-Tiny/ImageNet-100 setting, or revise the abstract, introduction, and conclusion so that the claims are explicitly limited to the setting studied in this paper.

**Non-critical**
- Please temper the detection discussion. Because the absolute COCO results are very low, the paper should clearly present those experiments as evidence about relative degradation trends rather than strong evidence about practical detection quality.

- An ablation for the ALiBi collapse at 96 px would further strengthen the paper. This is one of the most interesting observations in the paper, and testing slope magnitude, distance normalization, or CLS-token handling would make the discussion more convincing.

---

### Review · Reviewer_1nxb · 2026-07-08

**Summary Of Contributions:**

This work studies resolution extrapolation in ViT-Tiny models with different positional encodings: learned absolute embeddings, RoPE, YaRN, ALiBi, and FIRE. Models are trained at a fixed low resolution and evaluated at higher resolutions without target-resolution fine-tuning. The main idea is that additive attention-bias methods, especially FIRE and ALiBi, are generally more robust than rotation-based methods such as RoPE under large resolution extrapolation. This work also introduces attention concentration metrics to explain the observed performance gap and reports additional observations about ALiBi’s downscaling failure and YaRN’s limitations at very small base resolutions.

**Audience:**

Yes

**Audience Explanation:**

- The topic is timely and relevant. Resolution-flexible ViTs are important for efficient high-resolution inference, detection, and segmentation.

- The comparison is practically useful. Positional encodings for ViTs would likely benefit from seeing how RoPE, YaRN, ALiBi, FIRE, and absolute interpolation behave under resolution shifts.

- The paper reports interesting empirical findings. The strong performance of FIRE/ALiBi at high resolution, the collapse of RoPE, and the surprisingly competitive Simple ViT baseline are all useful observations.

- The attention concentration analysis is a useful diagnostic. Even if not fully causal, the proposed scale-normalized metrics may help future work analyze extrapolation behavior.

**Claims And Evidence:**

Yes

**Claims Explanation:**

Partially.

- The main classification claim is well supported. On ImageNet-100, FIRE and ALiBi clearly retain much higher accuracy than RoPE at large extrapolation scales, and YaRN substantially improves over raw RoPE.

- The dense-prediction evidence is weaker. COCO mAP values are extremely low, and the detection results are not uniformly favorable to bias-based methods. Thus, strong claims about detection/general dense-prediction robustness should be softened.

- The mechanistic explanation is suggestive but overclaimed. The attention entropy/fanout analysis is useful, but it shows correlation rather than proving that attention concentration causes extrapolation robustness.

- Some implementation/theoretical claims need clarification. The YaRN setup appears ambiguous because the paper says it uses the same checkpoint as RoPE, while also listing different frequency bases. The claim that the proposed 2D FIRE extension is uniquely faithful or strictly translation-invariant is also not fully justified.

**Requested Changes:**

- Clarify the YaRN/RoPE implementation and remove possible confounding. The paper states that YaRN uses the same checkpoint as RoPE but also lists different hyperparameters for RoPE and YaRN, e.g., RoPE with $\theta=10$ and YaRN with $\theta=10000$. This creates ambiguity about whether YaRN is truly an inference-time modification of the same trained RoPE model. The authors should clearly state several implementation details, such as: What RoPE frequency base is used during training; Whether the RoPE and YaRN results are produced from exactly the same trained checkpoint; What parameters are changed at YaRN inference time.

- Clarify the broad “bias-based methods outperform rotation-based methods” claim. The classification and segmentation results support this claim at high extrapolation, but the detection results are more mixed. For example, RoPE and YaRN achieve higher mAP around the training resolution, and YaRN has competitive AP50 at $1024\times 1024$. Moreover, absolute COCO mAP values are very low, making strong practical conclusions difficult. The authors should revise claims such as “across all configurations” or “dense prediction confirms” to more precise language, e.g., “bias-based methods show stronger high-resolution robustness in classification and segmentation, and more stable relative degradation in detection.” The abstract, introduction, and conclusion should all be adjusted accordingly.

- Weaken and better support the ALiBi interpolation/topological-failure explanation. The ALiBi downscaling collapse is interesting, but the current claim that it is a “fundamental topological flaw” is too strong. The presented ablations rule out only a few simple fixes: CLS distance scaling, slope scaling, and joint scaling. They do not prove that topology is the necessary cause. The authors should clarify the conclusion as an empirical hypothesis: ALiBi appears sensitive to grid shrinkage, possibly because smaller grids change edge/center token proportions and local neighborhood structure. It would help to add more evidence.

- Report limited seed variability or uncertainty. All main results appear to be single-seed experiments. Since some differences between methods are modest near the training resolution and in dense prediction, reporting at least limited seed variability would improve confidence. A full multi-seed study across all tasks should be added, or the paper should explicitly state that the reported numbers are single-seed and avoid overinterpreting small differences.

---

### Review · Reviewer_mhs5 · 2026-07-15

**Summary Of Contributions:**

The authors present a systematic comparison of positional encoding strategies for Vision Transformers (ViT). The underlying motivation is to be able to train computationally expensive ViT models at lower resolutions and maintain good predictive performance when extrapolating to higher resolutions at test time. The authors compare two “attention bias-based” methods (FIRE and ALiBi) to two “rotation-based” methods (RoPE and YaRN) with a simple bicubic interpolation method (Simple ViT) used as a baseline. They provide a brief methodological overview of each method and then analyze their relative performance on standard benchmark datasets for classification, detection, and segmentation tasks. They conclude that bias-based methods are generally more robust under extrapolation.

### Strengths

- The introduction and motivation for the work is clearly presented

- The premise of the study is sound and the basic experimental design is reasonable

- The results are suggestive of some potentially interesting discrepancies between positional encoding methods


### Weaknesses

- Lack of rigor in presentation of methods and results

- Overly verbose text with excessive use of undefined jargon

- Poor model performance on several tasks undermines credibility

- No formal quantification of uncertainty or variability in experimental results

**Audience:**

Yes

**Audience Explanation:**

The manuscript aims to provide a methodological comparison of extrapolation performance for state-of-the-art machine learning algorithms and is thus clearly within scope of TMLR.

**Claims And Evidence:**

No

**Claims Explanation:**

The study as presented lacks methodological rigor in at least four key ways:

1. **Representativeness of the models**. The authors use a relatively small transformer model (citing computational limitations) that performs relatively poorly on several of the  benchmark tasks; e.g. the mAP scores on the detection task are in the abysmal range of 1-13%. They base the validity of their results on the assumption that they expect “relative trends between positional encoding methods to generalize”, citing the “consistency” of their findings across tasks. However, as discussed further below, their findings are actually *not* very consistent across tasks, with substantial differences even within tasks (e.g. ImageNet vs. Tiny-ImageNet). No further citation or empirical basis is given for the assumption that the results will generalize.

2. **Misleading or unsupported claims**. The text currently contains numerous claims that either lack a firm theoretical or empirical basis or are inconsistent with the presented evidence. A few examples:

    - The authors claim that the FIRE normalizer $\psi(\max{L, i})$ guarantees the input to the MLP is bounded between 0 and 1, but fail to note that this assumption only holds for causal attention where $i - j < i$.

    - In Section 3.3, the authors claim that “…this is the **unique** **2D extension** with this property: the flattened and Euclidean variants each require modifications that have no 1D counterpart, while axis decomposition extends 1D FIRE symbol-for-symbol.” without any theoretical analysis or argument for uniqueness.

    - The authors draw the conclusion that bias-based methods generally out-perform rotation based methods but fail to acknowledge that, in their Tiny-ViT experiment (Figure 2), the bias-based ALiBi method under-performs both rotation-based methods.

    - Similarly, on the detection task, the authors point to the fact that the rotation-based methods show a steeper decline in performance, but seem to neglect the fact they also start from a higher baseline performance (for medium and large objects) and show similar performance at the highest resolution.

    - In Section 5.1, the authors claim that “In the near-training regime (224–384 px), several methods improve beyond their training-resolution accuracy, indicating that higher resolution inputs can provide genuinely useful additional spatial information when the positional encoding remains well behaved.” but seem to base this inference on very small apparent increases that are well within the initial spread of performance on the training resolution. Without any measures of uncertainty/variability in the performance metrics, it’s impossible to determine whether this finding is robust.

    - The caption for Figure 7 claims that the attention maps qualitatively confirm the authors’ hypothesis that bias-based methods “concentrate on the subject as resolution grows”, but this is inconsistent with what is actually shown. The bias-based FIRE method seems to (incorrectly) concentrate significant attention on the branch at both the training resolution and higher resolution, while the rotation-based RoPE method still shows bright spots on the bird even at high resolution.

3. **Lack of uncertainty quantification**. None of the authors quantitative results feature any measure of uncertainty or variability across multiple runs with different random seeds and train/test splits. This makes the relative performance differences between methods difficult to assess.

4. **Undefined jargon and evaluation metrics**. The manuscript is littered with undefined acronyms and technical jargon.

    - Despite being central to their objective, the authors never provide a precise definition for “true zero-shot extrapolation”.

    - Terms like “CLS-token” and “NTK-by-parts” are used without definition.

    - Evaluation metrics like mAP and mIoU are never defined.

**Requested Changes:**

The authors absolutely must address each of the key concerns flagged above. In addition, if the paper is to be published, the following deficiencies should also be addressed:

**Inconsistent notation**. The authors use inconsistent notation throughout their methodology section, e.g. $\circ$ vs. $\odot$ for the Hadamard product, $d$ as both the embedding dimensionality and distance,  both $\tilde{\mathbf{q}}, \tilde{\mathbf{k}}$ and $\bar{\mathbf{q}}, \bar{\mathbf{k}}$ for queries/keys, $m$ and $n$ vs. $i$ and $j$ for indices, and $\theta_t$ vs $\theta_d$ for the RoPE frequencies. These inconsistencies, if intentional, are not explained.

**Overly verbose and repetitive text**. A lot of the discussion in Sections 3 and 5 is repetitive and could be condensed.

**No code or data archive**. The authors do not provide any links to an archive of their code and data (though much of the latter should be publicly available) which hinders reproducibility and transparency.

**Bibliographic errors**. The manuscript contains several bibliographic errors, with incorrectly spelled or even entirely fabricated author names listed in several of the references:

> Xiangyu Hong, Jia Che, Biqing Jiang, Fandong Qi, and Yu Mo. On the token distance modeling ability of higher RoPE attention dimension. In Findings of EMNLP, 2024.

Incorrect author names, see https://aclanthology.org/2024.findings-emnlp.338/

> Xiaoran Liu, Yuerong Yin, Zhigeng Liu, and Fangming Huang. Beyond real: Imaginary extension of rotary position embeddings for long-context LLMs. arXiv preprint arXiv:2512.07525, 2025.

Incorrect author names, see https://arxiv.org/abs/2512.07525

> Jianlin Su, Yu Lu, Shengfeng Pan, Ahmed Murtadha, Bo Wen, and Yunfeng Liu. Roformer: Enhanced transformer with rotary position embedding. Neurocomputing, 568:127063, 2024.

Incorrect author order, see: https://www.sciencedirect.com/science/article/pii/S0925231223011864

> Qingyuan Tian, Zhijian Wenhong, Liaofeng Xiao, and Ru Wang. MrRoPE: Mixed-radix rotary position embedding. arXiv preprint arXiv:2601.22181, 2026.

Incorrect author names, see: https://arxiv.org/abs/2601.22181

> Tongyao Zhu, Qian Liu, Wang Haonan, Shiji Chen, and Min Xiang. SkyLadder: Better and faster pretraining via context window scheduling. In NeurIPS, 2025.

Incorrect author names, see: https://proceedings.neurips.cc/paper_files/paper/2025/hash/b837547467ea02deebdc9677d7f5a0be-Abstract-Conference.html